# Laser-Synthesized Elemental Boron Nanoparticles for Efficient Boron Neutron Capture Therapy

**DOI:** 10.3390/ijms242317088

**Published:** 2023-12-04

**Authors:** Irina N. Zavestovskaya, Anna I. Kasatova, Dmitry A. Kasatov, Julia S. Babkova, Ivan V. Zelepukin, Ksenya S. Kuzmina, Gleb V. Tikhonowski, Andrei I. Pastukhov, Kuder O. Aiyyzhy, Ekaterina V. Barmina, Anton A. Popov, Ivan A. Razumov, Evgenii L. Zavjalov, Maria S. Grigoryeva, Sergey M. Klimentov, Vladimir A. Ryabov, Sergey M. Deyev, Sergey Yu. Taskaev, Andrei V. Kabashin

**Affiliations:** 1P. N. Lebedev Physical Institute of the Russian Academy of Sciences, Moscow 119991, Russia; grigorevams@lebedev.ru (M.S.G.); ryabov@lebedev.ru (V.A.R.); 2Institute of Engineering Physics for Biomedicine (PhysBio), National Research Nuclear University MEPhI, Moscow 115409, Russiazelepukin@phystech.edu (I.V.Z.); gvtikhonovskii@mephi.ru (G.V.T.); aapopov1@mephi.ru (A.A.P.); smklimentov@mephi.ru (S.M.K.); biomem@mail.ru (S.M.D.); 3Laboratory of BNCT, Budker Institute of Nuclear Physics of the Siberian Branch of the Russian Academy of Sciences, Novosibirsk 630090, Russia; a.i.kasatova@inp.nsk.su (A.I.K.); d.a.kasatov@inp.nsk.su (D.A.K.); kuzminaksenya0102@mail.ru (K.S.K.); s.yu.taskaev@inp.nsk.su (S.Y.T.); 4Shemyakin-Ovchinnikov Institute of Bioorganic Chemistry of the Russian Academy of Sciences, Moscow 117997, Russia; 5LP3, Aix-Marseille University, CNRS, 13288 Marseille, France; andrei.pastukhov@etu.univ-amu.fr (A.I.P.); andrei.kabashin@univ-amu.fr (A.V.K.); 6A. M. Prokhorov General Physics Institute of the Russian Academy of Sciences, Moscow 119991, Russia; aiyyzhy@phystech.edu (K.O.A.); barminaev@gmail.com (E.V.B.); 7Institute of Cytology and Genetics of the Siberian Branch of the Russian Academy of Sciences, Novosibirsk 630090, Russia; razumov@bionet.nsc.ru (I.A.R.); zavjalov@bionet.nsc.ru (E.L.Z.); 8Laboratory of Molecular Pharmacology, Institute of Molecular Theranostics, Sechenov First Moscow State Medical University (Sechenov University), Moscow 119991, Russia; 9“Biomarker” Research Laboratory, Institute of Fundamental Medicine and Biology, Kazan Federal University, Kazan 420008, Russia

**Keywords:** boron neutron capture therapy, cancer treatment, boron nanoparticles, laser-ablative synthesis

## Abstract

Boron neutron capture therapy (BNCT) is one of the most appealing radiotherapy modalities, whose localization can be further improved by the employment of boron-containing nanoformulations, but the fabrication of biologically friendly, water-dispersible nanoparticles (NPs) with high boron content and favorable physicochemical characteristics still presents a great challenge. Here, we explore the use of elemental boron (B) NPs (BNPs) fabricated using the methods of pulsed laser ablation in liquids as sensitizers of BNCT. Depending on the conditions of laser-ablative synthesis, the used NPs were amorphous (a-BNPs) or partially crystallized (pc-BNPs) with a mean size of 20 nm or 50 nm, respectively. Both types of BNPs were functionalized with polyethylene glycol polymer to improve colloidal stability and biocompatibility. The NPs did not initiate any toxicity effects up to concentrations of 500 µg/mL, based on the results of MTT and clonogenic assay tests. The cells with BNPs incubated at a ^10^B concentration of 40 µg/mL were then irradiated with a thermal neutron beam for 30 min. We found that the presence of BNPs led to a radical enhancement in cancer cell death, namely a drop in colony forming capacity of SW-620 cells down to 12.6% and 1.6% for a-BNPs and pc-BNPs, respectively, while the relevant colony-forming capacity for U87 cells dropped down to 17%. The effect of cell irradiation by neutron beam uniquely was negligible under these conditions. Finally, to estimate the dose and regimes of irradiation for future BNCT in vivo tests, we studied the biodistribution of boron under intratumoral administration of BNPs in immunodeficient SCID mice and recorded excellent retention of boron in tumors. The obtained data unambiguously evidenced the effect of a neutron therapy enhancement, which can be attributed to efficient BNP-mediated generation of α-particles.

## 1. Introduction

Boron neutron capture therapy (BNCT) is a binary method of radiation therapy that is based on the nuclear capture and fission following the irradiation of nonradioactive boron-10 (^10^B) with low thermal neutrons (<0.5 eV), which leads to the production of an α particle and lithium-7 (^10^B + ^1^n → ^4^He(α) + ^7^Li + 2.79 MeV) with energy of 1.47 MeV and 0.84 MeV, respectively [1,2,3,4]. Capable of efficiently destroying the DNA of rapidly proliferating cancer cells [5], the generated α particles are characterized by high linear energy transfer and small path lengths of 5.2 and 7.5 μm, resulting in main energy deposition (84%) limited to the single tumor cell [4], which makes BNCT a selective method of cancer treatment with minor systemic side effects. 

First described by G. L. Locher in 1936 [6], BNCT has seen major development starting from the 1950s [1]. The first studies using first-generation boron-containing drugs borax and sodium pentaborate were accompanied by severe side effects [7], while later works employed more efficient drugs, better suited for biological systems, sodium borocaptate (BSH) and p-boronophenylalanine (BPA). As an example, A. H. Soloway [8] and Y. Nakagawa [9] reported high 5-year survival rates of patients with high-grade gliomas using BSH as a sensitizer in BNCT, while Y. Mishima [10] achieved promising results for the use of BPA in the treatment of patients with superficial malignant melanoma. BNCT has been the focus of numerous works with clinical studies performed by different groups in many countries, while indications for BNCT were extended to head and neck tumors, meningioma, hepatocellular carcinoma, liver metastases from colorectal cancer, breast cancer, pleural mesothelioma, and some other diseases [2,3,4].

Despite the remarkable success of BNCT technology, it is still considered an experimental method, which is not included in conventional cancer treatment schemes. One of the problems is related to the high cost of instruments, as most studies have been performed using epithermal neutron sources from bulky nuclear reactors. Furthermore, some reactors were closed for clinical investigations due to political, economic, or radiophobia reasons. However, the recent spectacular advancements in alternative neutron sources based on compact accelerators hold promise for a radical reduction in the cost of BNCT technology and its implementation in hospitals and clinical centers [4]. One such efficient compact source is based on an original design of a tandem accelerator (vacuum insulated tandem accelerator) and a solid lithium target was developed by the authors of this paper and installed at the Budker Institute of Nuclear Physics, Novosibirsk, Russia [11]. Such a source of neutron radiation has already been optimized and successfully tested in a variety of cancer treatment tasks and clinical trials [12,13]. Another major problem is related to the lack of efficient and safe systems to deliver ^10^B to cancer cells. BNCT protocols require a selective accumulation of ^10^B in tumor tissues at high concentrations (20–50 µg of ^10^B per 1 g of tissue), the retention of ^10^B in the tumor for several hours during irradiation, and quick clearance from blood and other organs [14]. For the moment, only second-generation drugs BPA and BSH are certified and used in clinical trials [2], but these drugs cannot satisfy these stringent requirements. Indeed, the accumulation of ^10^B in tumor tissues after BSH or BPA is not selective and strongly depends on the histological heterogeneity of patients [15,16,17], which requires the injection of high doses of these drugs leading to severe side effects. In addition, these drugs cannot ensure prolonged retention of boron in the tumor, while BPA is poorly water-dispersible [18].

High expectations in the advancement of BNCT are now related to the employment of nanotechnology approaches [19,20], which employ formulations of boron-containing nanoparticles (NPs) instead of BSH or BPA solutions. These approaches are known to offer a passive mechanism for targeting tumors based on enhanced permeability and retention (EPR) effect [21], and profit from additional therapeutic or imaging functionalities offered by NPs, as was shown earlier for a variety of inorganic nanomaterials (see, e.g., [22,23,24]). Several studies have already reported the use of boron-containing NPs, including distearoyl boron lipid (DSBL) liposomes [25], borophene nanomaterials [26], boron carbide NPs [27,28], boron nitride (BN) NPs [29] and nanotubes [30,31], ^10^B-enriched boron carbon oxynitride NPs [32], and ^10^B@SiO_2_@Gd NPs [33]. In particular, an in vivo study of boron carbide NPs in a B16-OVA mice model showed a highly significant increase in the median survival after neutron irradiation, while the boron uptake by tumor tissue was about 75–100% of the subcutaneously injected amount of boron [28]. Kuthala et al. [33] reported promising results using ^10^B-enriched (96% enrichment) ^10^B@SiO_2_@Gd NPs, which were surface-modified with FITC-labeled RGD-K peptide in a murine glioblastoma ALTS1C1 model. The authors reported a successful in vitro uptake via receptor-mediated endocytosis and a reduction in cellular viability after BNCT with this boron-containing agent. In vivo magnetic-resonance-imaging-guided BNCT showed tumor growth suppression and the prolongation of the average half-life time of mice from 22 to 39 days. In another representative study, Li et al. [29] used boron nitride NPs coated by a phase-transitioned lysozyme, which did not show significant cytotoxicity at boron concentrations up to 64 μg/mL and demonstrated a significant reduction in cell viability. In vivo neutron irradiation with BNPs on mice with 4T1 triple-negative breast cancer xenografts showed a 10-day delay in tumor growth, while the average tumor volume was four times lower compared to the control group for 20 days after the irradiation. 

The first data on the use of boron-containing nanoformulations look very encouraging, but subsequent advances in BNCT nanotechnology will require further optimization of nanomaterial designs to maximize boron content in cancer cells under a relative safety and biodegradability (in the ideal case) of used nanoformulations. Here, pure elemental boron NPs seem to be especially promising as they can deliver the maximal content of this element to cancer cells, but the synthesis of non-toxic, water-dispersible, and non-aggregated boron NPs presents a great challenge. Indeed, chemical synthesis routes typically provide NPs contaminated by toxic reaction byproducts [34,35], while NPs synthesized by dry fabrication methods such as laser pyrolysis, ultrasound, or mechanical milling typically have wide size distribution, and aggregate and tend to precipitate during their dispersion in aqueous solutions [36]. 

We believe that the above-stated stringent requirements for the design of BNCT-related boron-containing nanoformulations can be addressed by the employment of pulsed laser ablation in liquids (PLAL) methods [37,38,39]. Here, a solid target immersed in a pure liquid ambience is ablated by pulsed laser radiation to produce nanoclusters, which then coalesce to form colloidal NP solutions [40,41]. Such a technique can provide a series of advantages, including the cleanness of nanomaterials (e.g., it can be performed in deionized water or organic solutions), and natural liquid dispersibility in the absence of aggregation effects due to the formation of NPs in a liquid environment. We recently demonstrated the efficient synthesis of boron-containing nanoformulations, including Fe_2_B NPs [42] and elemental boron NPs (BNPs) [43,44], using PLAL methods. The ablation by long (nanosecond) laser radiation in organic solutions typically leads to the formation of partially carbonized amorphous boron NPs (a-BNPs) of small size (10–30 nm) [43], while ultrashort (femtosecond) laser ablation in deionized water could result in the formation of partially crystalline elemental BNPs (pc-BNPs) of slightly larger size (30–50 nm) and boric acid as a byproduct, which could be later removed by centrifugation [44]. We also showed that laser-synthesized BNPs represent promising sensitizers of photothermal therapy under IR irradiation [45], which could be enabled in parallel with a radiotherapy channel to increase therapeutic outcomes.

Here, we explore the use of elemental BNPs, synthesized using the method of pulsed laser ablation in liquids, as sensitizers of BNCT in vitro. We show that such NPs can indeed drastically enhance the therapeutic outcome, promising an advancement in BNCT technology. 

## 2. Results

### 2.1. Synthesis and Characterization of Boron NPs

In our experiments, we fabricated amorphous BNPs (a-BNPs) and partially crystalline BNPs (pc-BNPs) by the methods of nanosecond (ns) and femtosecond (fs) laser ablation, respectively, as described in detail in the Materials and Methods section. Briefly, a-BNPs were synthesized by the ablation of a pure boron micropowder by 200 ns laser radiation at the wavelength of 1060–1070 nm in isopropanol, followed by the laser fragmentation of the formed solution. For the formation of pc-BNPs, we used a technique of ultrashort (270 fs) laser ablation from a bulk boron target in deionized water, similar to our previous work [44]. In both cases, PLAL led to fast coloration of the liquid in the ablation chamber, indicating NP formation. The color was dark brown for the highly concentrated colloid. After NP formation, all solutions were subjected to a centrifugation step to remove large NPs.

Laser ablation and fragmentation of a boron micropowder with ns pulse duration resulted in the formation of NPs with a morphology close to spherical (Figure 1a). The size distribution was in the range of 25–30 nm, with a size dispersion of 17 nm full-width-at-half-maximum (FWHM) (see Figure 1a). Such BNPs presented amorphous elemental boron with a certain carbon contamination (Appendix A), due to a partial decomposition of isopropanol under its optical breakdown, as determined in our previous work [43]. On the other hand, fs PLAL of a bulk boron target in deionized water led to the formation of slightly larger NPs with a strictly spherical morphology (Figure 1b). As we showed earlier [44], fs laser-ablated BNPs were partially crystallized with the presence of a significant amorphous phase. The size distribution was about 50 nm with a FWHM of 65 nm (Figure 1b). 

### 2.2. Functionalization of Boron NPs

BNPs were coated with polyethylene glycol (PEG) to improve their colloidal stability. We used a modification of the Stober method, where a molar excess of 5 kDa mPEG-Silane was hydrolyzed and condensed on the particle surface. After the coating, PEG-a-BNPs had the mean hydrodynamic size of (70 ± 31) nm in water, while in phosphate-buffered saline (PBS), simulating physiological conditions, a slight increase in the hydrodynamic size up to (96 ± 46) nm was observed (Figure 1c). For PEG-pc-BNPs, the average hydrodynamic sizes in water and in PBS were (103 ± 37) nm and (121 ± 34) nm, respectively (Figure 1d). After the PEGylation, the BNPs had a strong negative surface charge with average zeta potentials of (–25 ± 6) mV for PEG-a-BNPs (Figure 1e) and (–34 ± 13) mV for PEG-pc-BNPs (Figure 1f).

The presence of mPEG-Silane molecules on the BNP surface was confirmed by energy-dispersive X-ray spectroscopy. Both uncoated pc-BNPs and PEG-pc-BNPs showed the presence of the K_α_ line of B element at 0.183 keV, as well as Mg and Al from the SEM wafer. The presence of an oxygen-related line in the unmodified particles can be attributed to the partial oxidation of BNPs or the oxidation of the Mg/Al wafer. In comparison with the unmodified BNPs, the EDS spectrum of the PEG-pc-BNPs demonstrated the appearance of the silicon (Si) elemental peak (Figure 1g), which confirmed the successful polymer attachment.

### 2.3. MTT Cytotoxicity Assay and Accumulation of Boron in Cells

The cytotoxicity of a-BNPs and pc-BNPs was evaluated via an MTT test after 24 h of cell cultivation in the presence of different concentrations of NPs. The MTT test follows the reduction of the tetrazolium dye by mitochondrial enzymes and monitors the metabolic activity of cells. Two human cancer cell lines were used for the experiments: glioblastoma U87 and colon adenocarcinoma SW-620. The cytotoxicity results are shown in Figure 2. One can see that a-BNPs demonstrated low cytotoxic levels up to concentrations of 400 µg/mL, while the tolerance of the U87 line was slightly higher (nevertheless, the viability of SW-620 cells was higher than 91.7% for the concentration of 400 µg/mL). pc-BNPs demonstrated a similarly low level of cytotoxicity. We did not observe a remarkable change in cell viability up to BNP concentrations of 200 µg/mL for the SW-620 and U87 cell lines (cell viability higher than 94%), while some cytotoxic effect was noted only at the BNP concentration of 800 µg/mL (cell viability less than 60% for the U87 cell line). The obtained results demonstrate the high biocompatibility of BNPs, prepared by laser ablation for the cancer cells. Based on the MTT assay, a 200 µg/mL dose of NPs was used for further experiments (equivalent to 40 µg/mL ^10^B dose). It should be noted that a similar ^10^B concentration was previously used in vitro for BNCT with a molecular compound of boronophenylalanine–fructose [46]. 

### 2.4. MTT-Based Evaluation of the Metabolic Activity of Cells after BNCT Treatment 

In our experiments, using the same MTT test, we compared the viability of cells to neutron beam exposure in the presence/absence of BNPs. The geometry of irradiation and the dose were optimized to avoid altering cell viability during the experiment [47]. The cells were irradiated by thermal neutron beam for 30 min, providing the equivalent calculated dose of 8 Gy-Eq, while ^10^B was used at a concentration of 40 µg/mL (200 µg/mL concentration of BNPs). MTT tests were used to compare cell viability in four groups: (i) cells after 4 days of exposure to BNPs; (ii) cells after irradiation with neutron beam alone; (iii) cells after irradiation with neutron beam in the presence of BNPs (BNCT conditions); (iv) control cell groups in the absence of NPs and neutron irradiation. The SW-620 cell line was used to compare the effect of a-BNPs and pc-BNPs, while U-87 cells were additionally tested for BNCT with pc-BNPs (Figure 3). 

SW-620 cells loaded with a-BNPs and subjected to neutron beam irradiation caused a significant reduction in cell viability, down to 67% compared to the control, irradiation, and NP-treated groups (Figure 3a). Even more remarkable results were recorded for the BNCT group incubated with pc-BNPs (Figure 3b). While our tests did not record any statistically significant differences between the NP-treated group compared to the control and irradiation-treated groups (cell viability was 94.7% and 96.6% for the U87 and SW-620 cell lines, respectively), the viability for the BNCT group decreased down to 42 and 47% for the U87 and SW-620 cell lines, respectively (Figure 3b). Such a result constitutes strong evidence for the involvement of the boron neutron capture mechanism associated with the generation of α-particles and their therapeutic action. 

### 2.5. Colony formation Assay to Estimate the Efficiency of BNCT Treatment

Since the metabolic activity of mitochondrial enzymes in cells can persist even after significant cell damage, we additionally evaluated cell toxicity using a clonogenic assay. Clonogenic tests show cell ability to divide and form colonies after the treatment. Since BNCT is likely to destroy the DNA of proliferating cancer cells by α particle generation, the clonogenic assay has become “a golden standard” for estimating toxicity in this field. 

The results of the colony formation assays for a-BNPs and pc-BNPs are shown in Figure 3c,d, respectively. One can see that the used epithermal neutron beam does not affect cell viability and proliferative capacity, since there were no statistically significant differences between the irradiation and control groups in both tests. Previous neutron beam experiments allowed us to optimize the geometry of this cell experiment by means of a PMMA phantom placed between the lithium target and the samples [48].

The results of the colony formation assay after NP exposure do not completely match the MTT data (Figure 3a,b). A minor reduction in cell metabolism in the group with a-BNPs did not affect the colony-forming capacity of cells (Figure 3c). The pc-BNP exposure was also relatively safe, reducing the survival fraction by only 4 and 6% for the SW-620 and U87 lines, respectively (Figure 3d). The BNCT group with amorphous boron NPs decreases the SW-620 colonies’ quantity to 12.6% (Figure 3c). The BNCT with crystalline boron NPs significantly decreased the survival fractions for both the U87 and SW-620 cell lines, which were 17 and 1.6%, respectively. These values show greater toxicity of BNCT to cells than MTT tests. The effect of boron neutron capture therapy with both boron NPs on metabolic activity and clonogenic capacity depended on the cell line because of different cellular radiosensitivity and boron distribution inside the cell [49,50]. 

Low cytotoxicity, sufficient uptake by tumor cells, and the effectiveness of BNCT with amorphous and partially crystalline boron NPs showed their potential applicability for further in vivo investigations, which are required to evaluate the possibility of using these agents in clinical practice. 

### 2.6. Biodistribution In Vivo 

One of the important issues in the use of boron-based compounds in BNCT tasks is related to the efficacy of retention of ^10^B in tumors. For the purpose of dose planning for BNCT in vivo experiments, which are scheduled as the next step, we carried out boron biodistribution tests in an immunodeficient SCID mice model under intratumoral administration of BNPs. The mice were implanted with heterotopic U87 tumors, while the biodistribution was assessed using ICP-AES. The time points were chosen in such a way as to determine what concentrations of boron would be found in the tumor and organs if irradiation began 30 min after the injection of NPs into the tumor and if the irradiation lasted 1 h. The results are shown in Figure 4. The highest accumulation of boron was identified in the tumor (56 µg/g and 82 µg/g at 30 and 90 min after pc-BNP administration, respectively). The concentration of boron in the blood was significantly lower and was 0.37 and 0.34 μg/mL after 30 and 90 min, respectively. The accumulation of boron in the surrounding normal tissue (skin and muscle) was at background levels at the 30 min time point and increased to 1 and 0.2 µg/g for skin and muscle, respectively, in 1.5 h. The same trend was found for the liver: boron concentration increased from 2.7 to 4.4 µg/g during the observation period. Background values were revealed in the spleen and brain at both time points. In the control group, boron concentrations in tumors and organs of interest were at background levels. We should note that concentration differences between the 30 min and 90 min datapoints were not statistically significant, indicating that BNP clearance and redistribution take longer time periods. 

It should be noted that those NPs contained natural boron, where the concentration of therapeutically suitable ^10^B is only 20%. The achieved boron accumulation in the tumor is sufficient for successful BNCT in the case of boron enrichment with the isotope ^10^B. According to the calculations, the equivalent dose in the tumor during BNCT would be 12 Gy-Eq, in the paw, 5.5 Gy-Eq, and the whole body of mice would receive 2.2 Gy-Eq. The dose value was estimated using Monte Carlo numerical simulation using the NMC code [51] and cross-sections from the ENDF-VII database and from the article [52]; the simulation code is described in detail in the article [53].

## 3. Discussion

Our experiments were performed using the U87 (human glioblastoma) and SW-620 (colorectal adenocarcinoma) cell lines, which are widely used for BNCT experiments [50,54,55]. Glioblastoma is a malignant brain tumor, characterized by a median survival time of 12–15 months and a 5-year relative survival of 5.5% [56]. Due to the poor prognosis, BNCT has been proposed as a treatment option for patients having this diagnosis, and clinical investigations have shown promising results [2,57]. Colorectal cancer is also ranked at leading positions in terms of prevalence and mortality, and its recurrence usually occurs within the first 2 years after removal of the primary tumor. BNCT may be an option for patients with colon and rectum tumors [58,59]. Although no clinical trials have been conducted on BNCT for colorectal cancer, a number of studies have been published on patients with colorectal liver metastases [60].

In our investigation, we studied the effectiveness of elemental BNPs, synthesized by PLAL methods, as a potential boron drug source for BNCT. We used two types of BNPs prepared under different conditions of laser-ablative synthesis: small (20 nm) amorphous nanoparticles (a-BNPs) and larger (50 nm) partially crystallized nanoparticles (pc-NPs). Our tests demonstrated the high efficiency of both types of BNPs in BNCT tasks. While 100 µg/mL of B or more are required for successful BNCT, our data did not reveal any sign of cytotoxicity at these concentrations of BNPs, while some cytotoxic effects were recorded only under high (>400 µg/mL) concentrations of BNPs.

Our study showed that the survival fraction in BNCT mediated by amorphous and partially crystalline boron NP groups was significantly lower than in the control and irradiation-only groups. The pc-BNPs showed slightly greater cytotoxicity for the SW-620 cell line after BNCT than a-BNPs. This difference was confirmed by both the MTT test and clonogenic assay results. Nevertheless, our data show that ^10^B content is much more important than the crystallinity of NPs and both amorphous and crystal phases can be efficiently used for BNCT.

Cytotoxicity with BNPs and irradiation at a 40 µg/mL ^10^B concentration was higher than for molecular compounds, reported before. Melanocytes and B16F10 melanoma cells were treated with BPA at a concentration of 220 µg ^10^B/mL and then irradiated at the IEA-R1 nuclear reactor (IPEN, Brazil). MTT results showed that the viability of B16F10 melanoma cells decreased to 40% and scanning electron microscopy revealed apoptotic bodies with degradation of the extracellular matrix [61]. The other in vitro BNCT BPA-mediated investigation was conducted at Nuclear Reactor RA-3 (Argentina) with a total absorbed radiation dose of 1.3 Gy [62]. Colony formation assay revealed survival of 20% and 4% for metastatic pigmented melanoma and primary amelanotic cell lines, respectively.

Boron uptake in vivo is an important parameter for determining the effectiveness of BNCT since it is the parameter that makes it possible to calculate the boron dose received by tumor cells during therapy. Moreover, based on our biodistribution data, the most reasonable timing for BNCT after drug administration, as well as the duration of irradiation, can be selected. The biodistribution of boron depends on the delivery drug, route of administration, and type of tumor [16]. Some scientists have highlighted the problem of insufficient boron accumulation by the tumor tissue and significant variability in boron concentrations in vitro, in tumor tissue in animal models, and also in patients [16,49,63]. Currently, a large amount of data on pharmacokinetics has been accumulated for two boron carriers of the 2nd generation: BPA and BSH [14]. Iwakura and co-workers studied boron biodistribution in mice bearing melanotic and amelanotic melanoma after BPA was administrated intraperitoneally and found that boron concentration in melanotic melanoma was 6.72 µg/g tissue while the amelanotic one reached 5.13 µg/g tissue and these boron values were higher than in the blood, brain, skin, and liver [64]. Garabalino et al. [59] studied the biodistribution of BSH injected intravenously at a dose of 50 mg ^10^B/kg in hamsters with oral cancer. Boron concentration in tumors ranged from 24 to 35 ppm in 3–10 h post-administration and decreased to 9 ppm at the time point of 12 h. The maximum tumor/normal pouch boron concentration ratio was 1.8. The authors also reported that boron contents in the blood, liver, and kidneys were higher than in the tumors [63].

The BPA accumulation mechanism is provided by system L of amino acid transporters and is due to more intensive proliferation and protein synthesis occurring in tumor cells [65]. BSH accumulation in the tumor occurs via passive diffusion through tumor-associated capillaries [66]. Nanoparticles can accumulate in tumor cells due to active processes of transcytosis [67] or enhanced permeability and retention effect [21]. Nevertheless, since in the median usually less than 1% of the injected dose of nanoparticles are accumulated in tumors [68], the intratumoral way of administration seems reasonable to avoid excessive uptake by macrophages. In addition, the introduction of nanoparticles directly into the tumor can reduce off-target toxicity.

It is important that nanoparticles can bind to the extracellular matrix in the tumor and evade rapid clearance from the tissue, in comparison to molecular boron-containing compounds. The quantity of particles in the tumor did not decrease within 90 min after the injection due to the decreased diffusion of the particles. Recent studies have shown that nanoparticle exit from tumors persists through intratumoral and peritumoral lymphatic vessels, but the process is greatly delayed for nanoparticles larger than 50 nm [69]. So, using pc-BNPs of (121 ± 34) nm hydrodynamic size for BNCT allows the retention of boron near the tumor cells for effective therapy.

In this study, we reached high boron concentrations in the tumor while in other organs and blood, the boron concentrations were lower. Nevertheless, we managed to achieve a boron concentration in the tumor of 82 µg/g with only 20% of the ^10^B isotope, which is a limitation of the study. It is reported that for effective BNCT, the content of ^10^B in tumor cells should reach 20–50 µg/g [70]. We assume that obtaining nanoparticles from a ^10^B-enriched target will allow us to perform successful BNCT in further studies. 

## 4. Materials and Methods

### 4.1. The Synthesis of Boron Nanoparticles

The synthesis of the BNPs was performed by ns and fs laser ablation methods in liquid ambience, which were thoroughly described in our previous studies [43,44]. The source of laser radiation for ns synthesis was an ytterbium-doped fiber laser with an average power of 20 W, a wavelength of 1060–1070 nm, a pulse repetition rate of 20 kHz, a pulse energy of 1 mJ, and a pulse duration of 200 ns. The lens focal length was 207 mm. Fragmentation time was 40 min. Boron micropowder suspensions in isopropanol (concentration 200 μg/mL) were irradiated with a laser scanning beam at a speed of 1 m/s entering through the glass window of the cell from below. In the case of fs PLAL, a bulk crystalline boron target was fixed vertically inside the glass cuvette filled with 50 mL of deionized water (18.2 MΩ/cm at 25 °C). The laser beam (3 mm diameter) of a Yb:KGW fs laser (TETA-10, Avesta, Russia, 1030 nm, 270 fs, 30 μJ, 200 kHz) was focused on the target surface by a 100 mm working distance F-theta lens. The thickness of the liquid layer between the inner glass wall and the boron target was 2.5 mm. The laser beam was moved over the target surface at a speed of 4 m/s by a galvanometric system (LScan-10, Ateko-TM, Ltd., Moscow, Russia) with a self-closed spiral scanning pattern (3 mm outside diameter). The duration of the ablation experiment was 60 min. The large (>150 nm) BNPs were removed from the obtained solution by centrifugation (13000 RCF, 1 min).

### 4.2. NP Characterization

The hydrodynamic size of the obtained BNPs was measured using the dynamic light scattering technique (Zetasizer Nano ZS, Malvern Instruments Ltd., Malvern, UK). Mode values of number-weighted size distributions were used as the hydrodynamic diameter. All measurements for size analysis were carried out in distilled water and in phosphate-buffered saline (pH 7.4). Hydrodynamic diameters describe mode values ± half-width of the peak of number-weighted size distributions. Smoluchowski approximation was used for ζ-potential calculation, and the measurement was performed in a 10 mM NaCl water solution.

The morphology and size of the synthesized boron NPs were characterized by means of a transmission electron microscopy (TEM) system (Carl Zeiss, Jena, Germany). Electron images were obtained at a 200 kV accelerating voltage. Samples for TEM measurements were prepared by dropping 1 µL of colloidal solutions onto carbon grids, which were subsequently dried at ambient conditions. The size distribution of the synthesized boron NPs was obtained by analysis of the TEM images in the ImageJ software (ver. 1.54f) using a circle fit approximation. The final distribution was based on measurements of 300 boron NP diameters.

Energy-dispersive X-ray spectroscopy (EDS) was obtained using an X-act EDS detector (Oxford Instruments, High Wycombe, UK) coupled with SEM. Samples for EDS measurements were prepared by dropping 2 µL of colloidal solutions onto an Al/Mg substrate, which was subsequently dried at ambient conditions.

### 4.3. Chemical Modification of BNPs

An amount of 1 mg of laser-synthesized BNPs was dispersed in 1 mL of 96% ethanol, and then 65 μL of distilled water was added. After 5 min of ultrasonication, 100 μL of 1 g/L 5 kDa mPEG-Silane solution in ethanol was quickly added to the BNP colloid under stirring to start the reaction of silane chain hydrolysis and condensation. Next, BNPs were heated at 60 °C for 2 h and further incubated at room temperature overnight. Then, NPs were washed 3 times with ethanol and distilled water via centrifugation at 10,000× *g* for 15 min.

### 4.4. Cells

Human tumor cell lines U87 (glioblastoma) and SW-620 (colorectal adenocarcinoma) were obtained from the ‘Center for Genetic Resources of Laboratory Animals’ of the Institute of Cytology and Genetics of the Russian Academy of Sciences (Novosibirsk, Russia). Cells were cultured in DMEM/F12 (1:1) medium (Capricorn-Scientific, Ebsdorfergrund, Germany) supplemented with 10% fetal bovine serum (HyClone, Logan, UT, USA) and the antibiotic kanamycin (Biochemist, Russia) and ciprofloxacin (Sintez, Dzerzhinsk, Russia) at 37 °C and 5% CO_2_ in culture flasks.

### 4.5. Cytotoxicity Assay

Cytotoxicity was verified according to the mitochondrial function which was evaluated with MTT (3-(4,5-dimethylthiazol-2-yl)-2,5-diphenyltetrazolium bromide). The MTT test is based on the ability of mitochondrial enzymes of living cells to reduce the MTT reagent into purple-blue intracellular formazan crystals that are soluble in dimethyl sulfoxide (DMSO).

Cells were seeded in 96-well plates at a density of 4 × 10^4^ cells per well in 100 μL of complete growth medium. After 24 h of incubation, the medium in the experimental wells was removed and the medium with BNPs at a BNP concentration of 50–800 µg/mL was added. The medium without drugs was added to the control groups and cultivated for 24 h at 37 °C. MTT 5 mg/mL was dissolved in 0.9% sodium chloride. At the end of the incubation period, the medium was removed and 100 µL of medium without serum and 10 µL of MTT solution (Dia-m, USA) were added to each well. The plates were incubated for 4 h at 37 °C and 5% CO_2_, after which the supernatant was replaced with dimethyl sulfoxide (DMSO) and resuspended until all formazan crystals were dissolved. Optical density was measured on a Multiskan SkyHigh spectrophotometer (Thermo, Waltham, MA, USA) at a wavelength of 595 nm. The data are presented as the percentage of viability in the experimental groups to the control of each cell line.

### 4.6. BNCT Experiment

Cells were cultivated in culture flasks; 200 µg/mL solution of NPs was added to the experimental samples 24 h before irradiation (the concentration of ^10^B was 40 µg/mL). The control groups and the irradiation groups were incubated without BNPs. Immediately before irradiation, the cells were removed with trypsin, counted, and centrifuged and a total of 5 × 10^5^ cells in 2 mL of growth medium were transferred into polypropylene cryotubes.

Irradiation was carried out at the accelerator-based neutron source at the Budker Institute of Nuclear Physics [11] in the horizontal part of the tract for 30 min at a beam energy of 2.0 MeV, and a current of 1 mA. Samples were placed below the lithium target in a PMMA phantom to a depth of 72 mm and a diameter of 50 mm. The equivalent calculated dose for irradiated samples was 8 Gy-Eq. Control samples were placed under the same conditions as irradiated ones but far from the radiation generation site.

### 4.7. Evaluation of the Metabolic Activity of Cells 4 Days after BNCT Using the MTT Test

Both cell lines were seeded in 96-well plates at a density of 2 × 10^4^ cells per well and incubated for 96 h under standard conditions, and then the MTT test was performed as described above.

### 4.8. Colony Formation Assay

Cells were seeded at a density of 200 cells per well in a 6-well plate containing 3 mL of complete growth medium. The plates were cultured in a CO_2_ incubator. After the number of cells in the control group colonies reached 50, the colonies were washed with phosphate-buffered saline (PBS), fixed with 10% formalin (Panreac AppliChem, Darmstadt, Germany), and stained with Giemsa solution (Sigma-Aldrich, St. Louis, MO, USA). Colonies were counted visually using a Zeiss Primo Vert light inverted microscope (Germany). The proportion of surviving cells was calculated using the formula:$$Survival\;fraction=\frac{plating\;efficiency\;in\;experimental\;group\;}{plating\;efficiency\;in\;control\;group}\times 100\%$$

### 4.9. Boron Biodistribution in U87 Bearing Mice

Immunodeficient SCID mice aged 8–12 weeks (SHO-PrkdcscidHrhr) were used in the boron biodistribution study. SCID males in the experiment were kept at the Center for Genetic Resources of Laboratory Animals ICG SB RAS (RFMEFI62119X0023) in individually ventilated cages (Tecniplast) by family groups of 2–4 individuals at ambient temperature (22–24 °C) and relative humidity of 30–60%, with a light/dark mode of 12/12. Food and water were provided ad libitum. Two weeks before the experiment, mice were subcutaneously injected with a U87 cell suspension (3 × 10^6^ cells in 100 μL) into the right hind paw. Dynamic observation and dimensional measurements of tumor xenograft were performed every 3 days. Tumor volume was calculated using the formula: $$Tumor\;volume=\left(tumor\;length\times {tumor\;width}^{2}\right)\times 0.52$$

For the study, 3 experimental groups were formed: (1) 30 min time point after NP injection (n = 3); (2) 90 min time point after NP injection (n = 3); (3) control group without drug injection (n = 3).

Elemental BNPs were administered intratumorally; the dose was 200 µg/mL of BNPs and depended on the volume of the tumor. At 30 and 90 min after the introduction of NPs, the planned euthanasia of animals was performed, and organs were collected. Tissue samples were dissolved with concentrated nitric acid and 30% hydrogen peroxide at 90 °C in a Dry Block Heater 2 system (IKA, Staufen, Germany). The boron concentration was measured by inductively coupled plasma atomic emission spectrometry (ICP AES) on a high-resolution spectrometer ICPE-9820 (Shimadzu, Kyoto, Japan). The device was calibrated using the Boron Standard for ICP (Sigma-Aldrich) in the range of 0.01–10 mg/L.

The boron concentration was measured using the formula:$$Boron\;concentration=\frac{measured\;concentration\times sample\;volume}{organ\;weight}$$

## 5. Conclusions

The obtained in vitro results using laser-synthesized amorphous and crystalline boron NPs revealed low cytotoxicity of the NPs and high effectiveness in BNCT tasks. Crystalline NPs demonstrated higher cell death during BNCT than amorphous ones. The irradiation parameters were chosen in such a way that they did not affect the viability and colony-forming capacity without boron-containing NPs. In vivo experiments showed high boron accumulation in tumor tissue and low concentrations in healthy organs and tissues after NP intratumoral injection. The limitation of the study was the usage of NPs containing natural boron, where the concentration of therapeutically suitable ^10^B isotope is only 20%. Production of boron-enriched NPs will further enable BNCT in vivo.

## Figures and Tables

**Figure 1 ijms-24-17088-f001:**
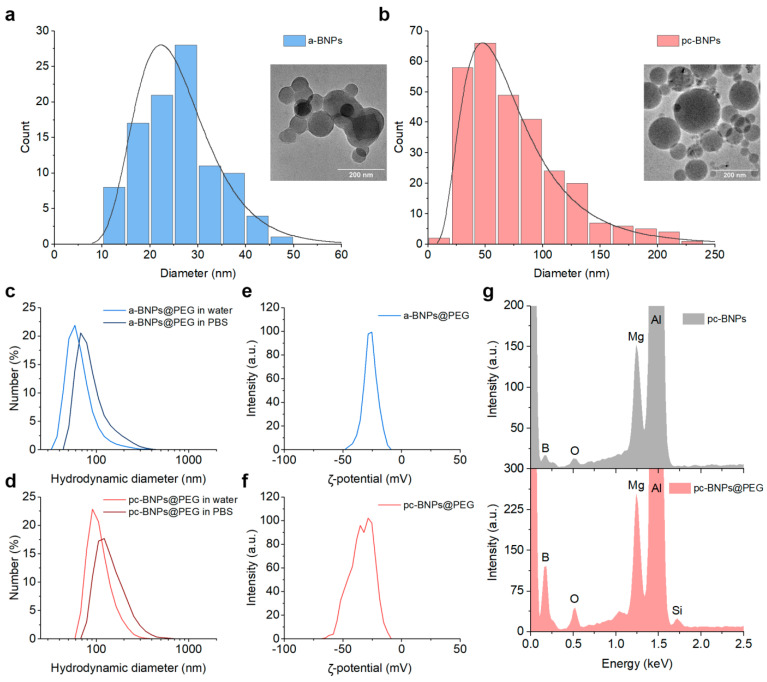
Representative TEM image and corresponding size distribution of a-BNPs (**a**) and pc-BNPs (**b**). Hydrodynamic size distribution of a-BNPs (**c**) and pc-BNPs (**d**) in water and phosphate-buffered saline (PBS). ζ-potential distribution of a-BNPs (**e**) and pc-BNPs (**f**). (**g**) EDS spectra of pc-BNPs before and after coating by mPEG-Silane. The main peaks of the elements are highlighted.

**Figure 2 ijms-24-17088-f002:**
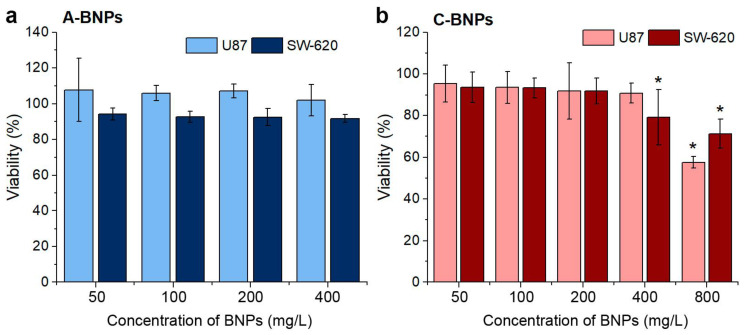
The viability of U87 and SW-620 cells exposed to increasing concentrations of a-BNPs (**a**) and pc-BNPs (**b**) after 24 h measured via an MTT test. The results are presented as mean ± SD. *—*p* < 0.05, compared to the control group, Mann–Whitney *U* test (n = 6).

**Figure 3 ijms-24-17088-f003:**
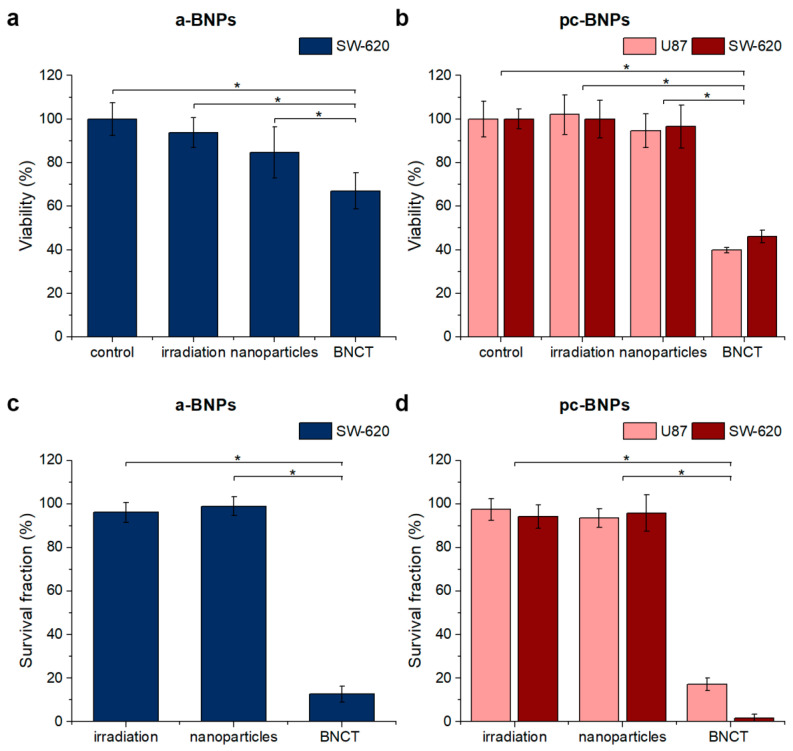
Results of MTT test 4 days after exposure (**a**,**b**) and colony formation assay 10 days after exposure (**c**,**d**) for a-BNPs (**a**,**c**) and pc-BNPs (**b**,**d**). The viability data represent the results after irradiation of cells alone, incubation of cells with laser-synthesized NPs, and BNCT (irradiation in the presence of NPs). The data are presented as mean ± SD. *—*p* < 0.05, differences between the control group and experimental groups, Mann–Whitney *U* test (n = 6).

**Figure 4 ijms-24-17088-f004:**
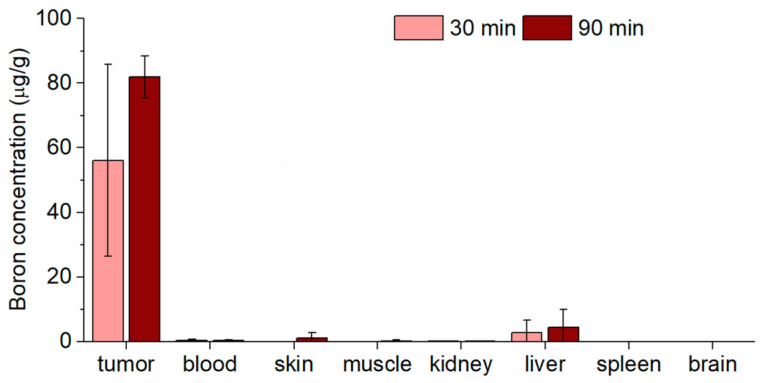
Biodistribution of pc-BNPs for 30 and 90 min after intratumoral administration. The data are presented as mean ± SD (n = 3).

## Data Availability

Data are available on reasonable request from the corresponding author.

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
