# Peer review of "Laser-Synthesized Elemental Boron Nanoparticles for Efficient Boron Neutron Capture Therapy"

_ijms, 2023, doi:10.3390/ijms242317088_

Round 1
Reviewer 1 Report
Comments and Suggestions for Authors
In this manuscript, the authors contribute to the advancement of BNCT as an alternative strategy for antitumor therapy by assessing the toxicity of elemental boron nanoparticles, synthesized by pulsed laser ablation liquid methods, their effectiveness for BNCT treatment (by MTT and clonogenic assays) and their accumulation in the tumor and organs of interest in a heterotopic mouse tumor model.
The article is well written and provides useful information for the improvement and application of BNCT. However, a few major and minor points should be addressed:
- The authors talk about the concentration of 10B in the different experiments (both in the figures and in the text), when they actually refer to the concentration of BNPs containing natural boron (where only about 20% is 10B), which leads to confusion. They should change this and indicate it correctly (BNPs concentration instead of 10B concentration) in figures and in the text.
- The results do not show the accumulation of boron in cells in vitro, although this is indicated in the abstract (line 36 “The NPs were efficiently absorbed by…”) and in the heading 2.3. of the results section (“MTT cytotoxicity assay and accumulation of boron in cells”). These results should be included; otherwise, the sentences referring to this determination should be deleted.
- Figure legend in figure 3 is not correct as it makes allusion to U87 cell line in figures a) and c) but these figures only show data from the SW-620 cell line. Why do they not show BNCT results of U87 with a-BNPs?
- Treatment times with BNPs to study cytotoxicity in cell lines should be longer, at least up to 48 hours to confirm that there is no long-term effect.
- Similarly, the times of BNPs biodistribution studied in vivo are very short. Although the results obtained can be extrapolated to a possible treatment 30 min after injecting the BNPs, it would be interesting to see their accumulation in the tumor and in other organs at longer times and to know how long it takes for them to be eliminated. Moreover, given that the concentration in tumor at 90 min is higher than at 30 min, it should be determined when the BNPs concentration in the tumor reaches its maximum value and is relatively stable, so that dosimetry can be more accurate. On the other hand, it would be interesting to analyze whether intravenous, rather than intratumoral, administration also leads to a greater accumulation of BNPs in the tumour than in other tissues.
- Since authors highlight the interest of BNCT for colorectal cancer in the manuscript, it would have been interesting to study in vivo biodistribution of BNPs in a tumor model by injection of SW-620 cells into nude mice.
- In the last paragraph of the results section (lines 305-307), an estimate of the equivalent dose in the tumour, paw and whole body of mice is given. How was this calculation performed?
- In line 287, it is figure 4 (not figure 5)
Author Response
In this manuscript, the authors contribute to the advancement of BNCT as an alternative strategy for antitumor therapy by assessing the toxicity of elemental boron nanoparticles, synthesized by pulsed laser ablation liquid methods, their effectiveness for BNCT treatment (by MTT and clonogenic assays) and their accumulation in the tumor and organs of interest in a heterotopic mouse tumor model. The article is well written and provides useful information for the improvement and application of BNCT. However, a few major and minor points should be addressed:
Response: We thank the reviewer for such a thorough review and numerous helpful comments and suggestions. We have taken all these comments and suggestions into account, and they have considerably improved our manuscript.
- The authors talk about the concentration of 10B in the different experiments (both in the figures and in the text), when they actually refer to the concentration of BNPs containing natural boron (where only about 20% is 10B), which leads to confusion. They should change this and indicate it correctly (BNPs concentration instead of 10B concentration) in figures and in the text.
In vitro data on nanoparticle toxicity and previously reported BNCT effects was referred to 10B concentration in the solution. We corrected text and figures, presenting BNPs concentration in bioavailability tests and in vivo assessment. For BNCT experiments we additionally mention 10B concentration as it is important for comparison to other studies.
- The results do not show the accumulation of boron in cells in vitro, although this is indicated in the abstract (line 36 “The NPs were efficiently absorbed by…”) and in the heading 2.3. of the results section (“MTT cytotoxicity assay and accumulation of boron in cells”). These results should be included; otherwise, the sentences referring to this determination should be deleted.
We have deleted accumulation of boron in cells in vitro in the Abstract, the heading 2.3. of the Results and Materials and Methods according to the Reviewer’s suggestions.
- Figure legend in figure 3 is not correct as it makes allusion to U87 cell line in figures a) and c) but these figures only show data from the SW-620 cell line. Why do they not show BNCT results of U87 with a-BNPs?
Please see the attachment
- Treatment times with BNPs to study cytotoxicity in cell lines should be longer, at least up to 48 hours to confirm that there is no long-term effect.
Please see the attachment
- Similarly, the times of BNPs biodistribution studied in vivo are very short. Although the results obtained can be extrapolated to a possible treatment 30 min after injecting the BNPs, it would be interesting to see their accumulation in the tumor and in other organs at longer times and to know how long it takes for them to be eliminated. Moreover, given that the concentration in tumor at 90 min is higher than at 30 min, it should be determined when the BNPs concentration in the tumor reaches its maximum value and is relatively stable, so that dosimetry can be more accurate. On the other hand, it would be interesting to analyze whether intravenous, rather than intratumoral, administration also leads to a greater accumulation of BNPs in the tumour than in other tissues.
This question is certainly very interesting. Pharmacokinetics of the drug depends on the route of administration. We will try to implement intravenous administration in our next studies.
In this work, the intratumoral route of administration was used due to its relative convenience and simplicity. Since this route of administration assumes that the concentration of nanoparticles (and, respectively, the concentration of boron) is maximum immediately after administration, however, for fixation of mice in restrictors and positioning under the neutron beam, time is required, which we defined as 30 minutes. The calculated time for irradiation takes 1 hour, so we chose time points of 30 and 90 minutes to determine the boron concentrations in the tumor and organs at the beginning of irradiation and at the end. We should note that concentration differences between 30-min and 90-min datapoints were non-statistically significant, indicating that BNPs clearance and redistribution takes longer time periods. This fact was commented in the text.
- Since authors highlight the interest of BNCT for colorectal cancer in the manuscript, it would have been interesting to study in vivo biodistribution of BNPs in a tumor model by injection of SW-620 cells into nude mice.
These experiments will be performed in further studies using 10B-enriched nanoparticles, along with BNCT treatment in vivo. Obtaining this data in current manuscript, describing in vitro effects is non-reasonable from the ethical point of view.
In current study we have shown the biodistribution of boron after intratumoral administration of nanoparticles only using a U87 bearing mouse model, since it is well known tumor model widely used for BNCT [1].
[1] Watanabe T, 2023, doi: 10.1007/s13318-023-00830-y; Zavjalov E, 2020, doi: 10.1080/09553002.2020.1761039; Alamón C, 2020, doi: 10.3390/cancers12113423; Coderre JA, 1990.
- In the last paragraph of the results section (lines 305-307), an estimate of the equivalent dose in the tumour, paw and whole body of mice is given. How was this calculation performed?
We we have corrected the Results section according to the Reviewer’s suggestions:
“The dose value is estimated using Monte Carlo numerical simulation using the NMC code [51] and cross sections from the ENDF-VII database and from the article [52]; the simulation code is described in detail in the article [53].”
- In line 287, it is figure 4 (not figure 5)
We have corrected the phrase according to the Reviewer’s suggestions.

Reviewer 2 Report
Comments and Suggestions for Authors
In this study, the authors try to explore the use of elemental BNPs, synthesized by the methods of pulsed 151 laser ablation liquids, as sensitizers of BNCT in vitro. This is a very promising study contributing to the narrow range of treatment options, especially for patients with human glioblastoma, when BNCT is proposed. The introduction is very comprehensive and covers all the aspects of this topic introducing the reader to the background of the Boron neutron capture therapy field. Very nice presentation of the results. In conclusion, a very novel and well-designed study that will raise the readers' interest.
Author Response
We thank the Reviewer for such a thorough review and recognition of our efforts in this study.
Round 2
Reviewer 1 Report
Comments and Suggestions for Authors
The comments have been addressed. The authors have included additional information that clarifies the issues and improves the quality of the paper.